# Object permanence in domestic cats (*Felis catus*) using violation-of-expectancy by owner and stranger

**Jemma Forman** [ORCID]*, **Jordan S. Rowe** [ORCID], **David A. Leavens**

School of Psychology, University of Sussex, East Sussex, United Kingdom

* j.forman@sussex.ac.uk

## Abstract

Object permanence, the understanding that objects continue to exist while out of sight, is a key part of the sensorimotor stage of cognitive development. Cats have been shown to reach Stage 5 object permanence by passing successive visible displacement tests, but their understanding of Stage 6 object permanence is less clear. We tested 18 domestic cats on their understanding of Stage 4 and Stage 6 of object permanence in their home environment. Additionally, we investigated how person familiarity may influence study engagement. In single visible displacement (SVD) tests, the box manipulator (owner or researcher) hid a toy in one of two cardboard boxes for the cat to find. In invisible displacement (IVD) tests, we implemented a violation-of-expectancy procedure in which the box manipulator showed a toy re-appearing out of either a) the same box it was hidden in (consistent trials) or b) the box it was not hidden in (violation trials). Approximately half of the cats (56%) did not find the hidden toy in SVD trials, with 42% of these cats not attempting to find the toy, despite previous research demonstrating that cats can retrieve hidden objects in successive SVD tests. None of our predictors significantly influenced whether cats found the toy, or which box was checked first (toy or empty box). In IVD trials, we unexpectedly found that cats were more likely to play with the toy and displayed more toy box-directed behaviours in consistent trials than violation trials. Similarly, we found that cats were more likely to display box-directed behaviours in trials where the researcher acted as the box manipulator. Breed, outdoor access, cat sex, and the first person to act as the box manipulator also influenced toy-directed behaviours. We discuss the complexity of person familiarity in research contexts and highlight some methodological challenges in studying cat cognition.

## Introduction

Object permanence is broadly defined as the understanding that objects continue to exist even when hidden out of sight [1,2]. Object permanence has six distinct stages

**Data availability statement:** All supplementary and data files are available from the figshare database (https://doi.org/10.25377/sussex.28287896).

**Funding:** The author(s) received no specific funding for this work.

**Competing interests:** The authors have declared that no competing interests exist.

that are developed within the sensorimotor period of a human infant [1]. In Stages 1 and 2, an infant does not search for a completely hidden object, but in Stage 3 can retrieve a partially hidden object. In Stage 4, an infant can successfully find an object which was overtly hidden in one hiding location during a single visible displacement (SVD) test. In Stage 5, an infant is capable of finding an object that has been moved in succession during successive visible displacement tests. In the final stage, Stage 6, an infant can successfully find an object that was covertly moved from one hiding location to another, normally within a separate container or behind an opaque barrier, during an invisible displacement (IVD) test. The development and milestones of object permanence is well-established in human infants, who display evidence of Stage 6 object permanence between 18 and 24 months of age [1,3,4].

Understanding of Stage 4 (single visible displacement) and 5 object permanence (successive visible displacements) has been evidenced in multiple adult animal species, such as primates [5–7], birds [8–10], ungulates [11,12], and marine mammals [13], as well as companion animals such as domestic dogs (*Canis familiaris*, [14–17]) and domestic cats (*Felis catus*, [18–22]). Object permanence has many inherent benefits for the survival of an animal in relation to locating food, avoiding predators, interacting with other members of their species and general navigation of their environment [23]. Furthermore, object permanence is likely beneficial for predators whose prey may become temporarily out of sight during a hunting sequence [24].

However, a critical perspective on animals' success in invisible displacement tasks suggested that great apes are the only species who demonstrate compelling evidence for understanding Stage 6 object permanence when controlling for sensory or social cueing and associative learning [25,26], although this has been debated and re-visited for bird species [27,28]. In domestic cats, Dumas [29] implemented an ecologically relevant procedure that took advantage of cats' hunting patterns and concluded that cats were able to correctly search for an invisibly displaced object, although this procedure has received similar criticism that associative learning was not disconfirmed [24]. Similarly, cats have been shown to correctly search for an apparently invisibly displaced food reward, yet not a toy reward [22], but associative learning strategies in this testing context were not controlled for the *last indicated* rule, of which the toy was always in the last location where the cover was manipulated [25]. Nonetheless, most studies have concluded that cats do not demonstrate evidence of understanding invisible displacement tasks [18,20,24,30]. Given that cats are skilled hunters that prey on multiple species that are capable of running, crawling or flying out of sight while being pursued by the cat [31–34], yet their understanding of IVD is not reliably evidenced in experimental settings, suggests that cats require sufficient incentivisation that complements their repertoire of hunting behaviours to accurately assess understanding of IVD tasks.

In our study, we aimed to address some of these methodological concerns and presented domestic cats with a violation-of-expectancy paradigm in an attempt to fairly assess their understanding of Stage 6 object permanence. In violation-of-expectancy paradigms [35,36], the subject is presented with an event that violates existing beliefs, for example seeing a screen seemingly rotate through a solid box or finding

a different object from the one that was seen to be hidden [35,37–40]. Upon seeing a violation event, in comparison to a knowledge-consistent event, the subject may display behaviours that are considered to be indicators of "surprise", such as an increase in: gaze duration towards the violation event [35]; pupil dilation [41–43]; freezing or stilling behaviours [44,45]; physical exploration of the object that violated expectations [46–48]; and gaze frequency towards caregivers [49]. From these behaviours, researchers have inferred that the subject has a mental representation of the object and may even attempt search for an explanation for the violation event, and thus have fully developed object permanence.

In addition to assessing object permanence, we investigated how the familiarity of the person carrying out the trials (the owner versus the researcher) may have impacted the behaviour of the cats during the study. The mere presence of an unfamiliar researcher in a cat's home environment may elicit a stress response in some cats, even if the researcher is not the one enacting the trial. This stress may manifest as passive behaviours such as freezing or crouching [50,51], so stilling behaviours or lack of physical exploration of objects may not be interpreted in the same way for cats in research settings in comparison to other animals such as dogs [48]. Overall, there is mixed evidence of how person familiarity may influence cat behaviour [52–58], which has important implications in assessing performance in experimental settings and interpreting cat cognition. Otherwise, the individuality of the cat has also been shown to be a significant contributing factor as to how a cat may behave with an unfamiliar person in relation to, for example, approaches, withdrawals, sniffing and distance between the cat and the unfamiliar person [53,59].

In the present study, we assessed domestic cats' understanding of Stage 4 (SVD) and Stage 6 (IVD) of object permanence using a variation of the violation-of-expectancy procedure in their home environment presented by both the owner (familiar person) and the researcher (unfamiliar person). For SVD trials, we overtly hid a toy in one of two cardboard boxes. As a result of increased stress from an unfamiliar person, we expected that when the researcher acted as the box manipulator, cats would be less accurate in their search behaviour and would find the toy less often in comparison to when the owner acted as the box manipulator. For event trials, we presented cats with a toy a) appearing out of the same box it was shown to be hidden (knowledge-consistent event) and b) appearing out of an adjacent box that it was not initially hidden in (violation event). In violation events, we expected that cats would have longer gaze durations at the toy, longer gaze durations at the manipulator, and more instances of touching the toy, in comparison to consistent events, similar to human infants and dogs [35,46–48]. Similar to SVD trials, as a result of person familiarity, we expected that when the researcher acted as the box manipulator, the cat would: play with the toy less often; have a longer latency to move towards the apparatus and play with the toy; and have lower frequencies of box-directed behaviours, in comparison to when the owner acted as the box manipulator. In both SVD and event trials, when the researcher acted as the box manipulator, we expected cats to have shorter gaze durations at the toy itself and longer gaze durations at the manipulator, in comparison to when their owner acted as the box manipulator. Furthermore, we also carried out exploratory analyses for whether demographic factors (such as cat age, cat sex, living in a multi-cat household, outdoor access, or breed) influenced any observed behaviours during SVD and event trials.

## Methods

### Ethics statement

Owners provided informed consent for their participation in this study. Approval was obtained by the Animal Welfare and Ethical Review Body at the University of Sussex (reference: ARG-29). The procedures used in this study were in accordance with the ethical standards of the institution guidelines.

### Pilot study

We tested three cats on a pilot procedure for this study. In this pilot procedure, a standalone, transparent barrier (120 cm x 65 cm) was placed in between the cat and two cardboard boxes. The boxes were approx. 50 cm apart from each other and

were placed approx. 10 cm from one side of the barrier. The cat was positioned approx. 150 cm away from the other side of the barrier. The researcher would show a reward to the cat (a ball of tin foil) and place it in one of the two boxes and swap the position of the boxes above the transparent barrier (SVD) or above the opaque barrier (IVD; a piece of cardboard was placed in front of the transparent barrier to become opaque). We found that cats could not navigate around the transparent barrier and instead walked directly towards the barrier in an attempt to reach the boxes (cats often fail transparent detour tasks, [60,61]). At this stage, the cat would sit in front of the barrier and not make another search attempt. Furthermore, as the apparatus was quite large, placing an opaque cardboard barrier in front of the transparent barrier often scared the cats away from the testing area.

We tested a fourth cat on a new procedure that did not involve the large barrier at all. Instead, we next attempted to pick up the cat and turn around while the boxes were being swapped. We also tested a procedure where the non-interacting person (either the researcher or the owner) sat between the boxes and the cat to obstruct the cat's view while the boxes were being swapped by the box manipulator. These methods were also too disruptive and stressful for the cat to participate in the study. Finally, we tested a revised method of using an A4 folder to temporarily obstruct the view of the boxes from the cat and found the cat was not apparently fearful of the smaller barrier. Despite no longer being fearful and running away, the cat was still hesitant to walk around the boxes to find the toy in IVD trials. As such, we no longer tasked cats with moving around the boxes in IVD trials and instead we presented a violation-of-expectancy paradigm to the cats to assess understanding of Stage 6 object permanence (final method described in *Procedure*). None of the cats from the pilot study were re-visited for participation in the main study.

## Subjects

Twenty-five cats from 18 households in Brighton, United Kingdom were initially recruited for this study. Four cats belonging to the same household were excluded; two of these cats walked away from the study area and could not be encouraged to participate, and two cats were not present at the time of the study. One cat was excluded for lack of motivation to engage with the study and two cats were excluded due to displaying fearful behaviour in the presence of the researcher (actively moving away from the researcher and hiding behind furniture).

Eighteen cats from 14 households were retained in the final sample. Ten cats lived in dual-cat households and eight cats lived in single-cat households. Four households had two participating cats each and two participating cats lived in separate households with one other non-participating cat.

## Apparatus

We presented cats with a selection of toys, including balls (approx. 4 cm in diameter), hair ties, small mice (approx. 5 cm in length), large mice (approx. 12 cm in length) and novelty cacti (approx. 10 cm in length) and encouraged the cats to play with them. Most cats preferred to play with a cactus toy ($n = 11$), followed by a large mouse toy ($n = 3$), small mouse toy ($n = 2$) and a hair tie ($n = 2$). All of the toys were new to each cat and did not contain catnip. Cats were always supervised when playing with the small toys (including hair ties) to prevent accidental swallowing. The toy that the cat displayed the most interest in (for example, physically playing with, sniffing at, or sitting in close proximity with) was used in subsequent experimental trials and the remaining toys were removed out of sight of the cat. Two cardboard boxes (16 cm x 16 cm x 16 cm each) were used to hide the preferred toy. The two boxes were instructed to be positioned next to each other and to be separated by a small distance (approx. 2 cm). The researchers glued low density foam (1/2" thickness) to the inside of the boxes to help to reduce sound cues from the toy being placed inside. An A4 folder acted as a visual block so cats could not see the boxes being manipulated during consistent and violation event trials. A wide-angle lens camera (Panasonic HC-X920) was set up next to the manipulating person and facing the inside of the boxes (Fig 1a) and a second camera (Sony DCR-SR37) was set up facing the outside of the boxes (Fig 1b).

a

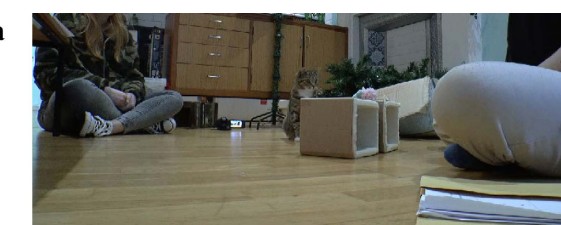

b

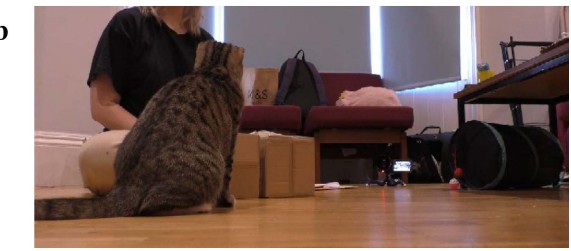

**Fig 1. Screenshots of the experimental setup in one owner's home during a violation trial.** The first camera was positioned facing the inside of the boxes **(a)** and a second camera was positioned facing the outside of the boxes **(b).** The manipulator (owner) is sat behind the open side of the boxes. The bystander (researcher) is sat behind the outside of the boxes. The cat is gazing at the toy placed on top of one of the boxes.

## Procedure

Owners were provided an information sheet and consent form to complete and sign prior to video recording. Owners were asked to complete a brief questionnaire that asked demographic questions on cat age, breed, sex, neuter status, adoption status, outside access and whether the cat lives in a multi-cat household. The study area was a place in the household that owners considered the cats to be the most comfortable (for example, a living room or bedroom). Participating cats from the same household were tested either individually or in the presence of the other resident cat, whichever the participating cat seemed most comfortable with. The researcher (adult female) was unfamiliar to all cats.

The movement of furniture, for example tables or chairs, was kept to a minimum to reduce disruption to the cat's home environment. The cat had a short period of time (5 mins) to explore the video cameras and task materials once the equipment was set in place. The researcher and the owner remained attentive towards the cat during the trials. In between trials, the owner would interact with the cat as normal in terms of talking, looking, petting or playing with the cat. The researcher maintained a friendly presence but largely ignored the cat unless they approached the researcher of their own volition. During the trials, the owner and the researcher were permitted to talk to each other, and the cat was allowed to move freely during the trials. Before the trials, the researcher physically demonstrated to the owner how they should carry out the study in relation to moving the boxes, the toy and the folder. The researcher also verbally instructed the owner what to do during the trials.

## Single visible displacement trials

The box manipulator would sit approximately 16 cm (one box length) behind the open side of the cardboard boxes; the open side of the boxes would face the manipulating person. The bystander would sit elsewhere in the same room either on the floor or in an available chair. Depending on the layout and size of the room, the bystander would sit between 1 m to 3 m away from the apparatus. The bystander and the box manipulator would always swap positions, so the bystander was always in the same place for every trial. Before every trial, the cat would have the opportunity to play with their preferred toy for a brief period of time (approx. 10 s) to encourage cats to be sufficiently motivated for participating in subsequent trials. The owner would call the cat's name to encourage the cat to look at the toy while the owner hides the toy in one of

the boxes. The first trial would start immediately after the cat had displayed substantial interest in its preferred toy. The cat would see the toy get placed inside either the left or right box (from the perspective of the box manipulator) at random. The manipulating person then sat for 60 s and waited for the trial to end. During this time, the manipulating person was attentive to the cat (i.e., facing and looking towards the cat) and was allowed to speak to the cat to encourage their participation (e.g., "Where's the toy?").

### Knowledge-consistent event trials

Knowledge-consistent trials initially followed the same procedure as visible displacement trials. However, once the toy was hidden in one of the boxes, the manipulator would slowly place the folder between the cat and the boxes so the cat could no longer see the boxes. The folder was moved slowly to avoid startling the cats as much as possible. The manipulator would then pretend to swap the boxes around by gently shaking both boxes for a few seconds to maintain consistent sound cues while the boxes were out of sight of the cat. Next, the manipulator slowly removed the folder and called the cat's name once again to get their attention to ensure they were looking in the direction of the boxes. The manipulator then removed the toy from the box and placed it on top of the same box it was removed from. The manipulator then waited 30 s in an attentive state for the trial to end. Once the trial had ended, if the cat had played with the toy, the closest person (either the manipulator or bystander) would retrieve the toy in preparation for the subsequent event trial.

### Violation event trials

Violation trials followed the same procedure as consistent trials. However, the box manipulator legitimately swapped the position of the boxes instead of pretending to swap them. From the cat's perspective, the consistent and violation trials appeared the same, but the location of the toy would either appear out of the expected box (consistent trials) or unexpected box (violation trials).

All 18 cats participated in 6 trials each across 108 trials. There were 36 SVD trials in total (18 with the owner and 18 with the researcher). There were 72 event trials in total (36 consistent events and 36 violation events). Similar to SVD trials, there were 18 consistent events and 18 violation events with the owner, and there were 18 consistent events and 18 violation events with the researcher. The first person to act as the box manipulator was counterbalanced (nine cats each). SVD trials were always presented first, followed by one consistent and one violation trial presented at random. Ten cats saw the consistent event first and eight cats saw the violation event first.

### Behaviour coding

All behavioural coding was carried out using the coding scheme in Table 1 using BORIS (Version 8.16.5, [62]). The observation interval started as soon as the manipulating person's hand moved away from the boxes after placing the toy. The observation interval ended once the allocated time had elapsed (60 s for SVD trials; 30 s for event trials) or if the cat pushed and followed the toy out of the observation area, whichever came first.

All videos were coded by one main coder (JF) and 22 trials were coded by a secondary coder (JR). There was excellent agreement for latency to first approach the apparatus (ICC = .92), latency to manipulate the toy (ICC = 1) and total duration of gazing at the box manipulator (ICC = .92). There was moderate agreement for total duration of gazing at the toy itself (ICC = .61). There were poor correlations for gazing at the bystander (ICC = .48) and toy investigation (ICC = .26) and these two variables were excluded from subsequent analysis. There were strong correlations between the two raters for the frequency of toy box approaches (Spearman's rho, $r = .84$) and empty box touches (Spearman's rho, $r = .78$). There were moderate correlations for the toy box touches (Spearman's rho, $r = .62$) and empty box approaches (Spearman's rho, $r = .66$). There were perfect correlations for whether the cat found the toy in SVD trials, the box that was checked first

**Table 1. Coding scheme of box-, toy- and person-directed behaviours displayed by the cat during single visible displacement (SVD) and event trials.**

| | Behaviour | Description |
|---|---|---|
| **Box-directed behaviour** | Approaches toy box | *Frequency* number of times the cat makes any movement with the head, body or paw towards the box the toy is hidden in or emerges from. This includes movement towards the outside or inside of the toy box. |
| | Approaches empty box | *Frequency* number of times the cat makes any movement with the head, body or paw towards the box the toy is not hidden in or does not emerge from. This includes movement towards the outside or inside of the empty box. |
| | Physical contact with toy box | *Frequency* the number of times the cat makes physical contact with the toy box (e.g., pawing at, jumping in, rubbing against the toy box). |
| | Physical contact with empty box | *Frequency* the number of times the cat makes physical contact with the empty box (e.g., pawing at, jumping in, rubbing against the empty box). |
| | Latency to first approach the apparatus | *Latency* time elapsed since the start of the observation interval until the first movement towards either the toy box or empty box. |
| | Found toy (SVD trials) | *Categorical* whether the cat found the hidden toy (found/not found/no attempt). The toy was 'found' if the cat gazed into the toy box, even if the cat did not touch the toy. The toy was 'not found' if the cat still made an attempt to find the toy (e.g., moving towards the boxes, gazing into the empty box) but did not gaze into the toy box. A trial was categorised as 'no attempt' if the cat made no movement towards the boxes. |
| | Box checked first (SVD trials) | *Categorical* the box which the cat gazed into first (toy box/empty box/no attempt). A trial was categorised as 'no attempt' if the cat made no movement towards the boxes. |
| **Toy-directed behaviour** | Plays with toy (event trials) | *Categorical* whether a cat makes physical contact with the toy (contact/no contact). *Latency* time elapsed since the start of the observation interval until the cat first makes physical contact with the toy. |
| | Investigate toy | *Duration* total duration of time the cat is sniffing (but not actively playing with) the toy. |
| | Total gaze at toy | *Duration* total duration of time the cat is gazing at the toy. |
| **Person-directed behaviour** | Gaze at box manipulator | *Duration* total gaze duration towards the person moving the toy and the boxes in the current trial. |
| | Gaze at bystander | *Duration* total gaze duration towards the person who is not moving the toy and the boxes in the current trial. |

during SVD trials, and whether the cat played with the toy in event trials (Cohen's Kappa: κ = 1, 100% agreement). The coding carried out by the primary coder was used in subsequent analyses.

The starting distance of the cat was defined by how many box lengths (16 cm) they were away from the nearest box at the start of the observation interval. Starting distances were categorised as approximately 0 cm (positioned directly next to or already touching the box), 16 cm (one box away), 32 cm (two boxes away) or ≥ 48 cm away (three boxes or more away).

## Data analysis

All analyses were carried out in R (Version 2023.06.2) using the broom, buildmer, car, effectsize, emmeans, EnvStats, detectseparation, ggpubr, gtsummary, lmerTest, MASS, performance and tidyverse packages. R code is available online at (https://doi.org/10.25377/sussex.28287896). For both SVD and event trials, the total durations of gazing at the toy and gazing at the box manipulator were converted into relative durations of time (proportion of time the behaviour was observed in a given trial; behaviour/ observation interval) to account for the different trial durations. Additionally, because of the low overall frequency of box-directed behaviours, toy box touches and approaches were summed together (toy box-directed behaviour), and empty box touches and approaches were summed together (empty box-directed behaviour). For both SVD and event trials, we carried out exploratory analyses for numerous lifestyle factors (cat age, cat sex, multi-cat household, outdoor access, and breed) and their effects on our outcome variables.

## Regression models

Linear mixed models were fitted for the relative duration of gazing at manipulator (SVD and event trials); relative duration of gazing at the toy (SVD and event trials); latency to move towards apparatus (event trials); and latency to play with the toy (event trials). Full models included nine fixed effects of: person trial (owner/researcher); first person condition (owner/researcher); toy hiding box (left/right); cat age; cat sex; whether cats lived in a dual-cat household (single/dual); whether cats had outdoor access (indoor-only/outdoor access); breed (mixed/purebred); and starting distance (0 cm, 16 cm, 32 cm or ≥ 48 cm). Full models also included a random effect of cat identity nested within household identity to account for multiple participating cats from the same household. Full models for event trials were identical to SVD trials but with two additional fixed effects of event type (violation/consistent) and the first event condition the cats were presented with.

Full model fit was assessed by visualising residuals and checking model convergence, singularity, multicollinearity (Variance Inflation Factor of more than 5) and outliers (using Cook's distance). All full models had issues with singularity to which random effects were removed and singularity issues were resolved. Final model selection was based on the most parsimonious model fit that balanced Type 1 error and statistical power without evidence of singularity or overfitting to allow for reliable interpretation [63]. A backward elimination process was used in which variables were removed from the full model one at a time and model comparisons were assessed using AIC (smaller AIC values preferred). Likelihood Ratio Tests were used to compare goodness-of-fit between full models and backward elimination final models. Variable removal stopped once the model fit did not show improvement in AIC from the previous model. Final models were re-assessed to ensure suitable model fit. Effect sizes were calculated using generalized eta squared ($\eta^2_G$). Effect sizes were interpreted as small (>.02 $x$ <.13), medium (>.13 $x$ <.26), or large ($x$ >.26) [64,65].

## Logistic models

Logistic models were fitted for: whether the cat found the hidden toy (SVD trials); which box the cat checked first (SVD trials); and whether the cat played with the toy (event trials). Full models had the same fixed and random effects as outlined in *Regression models*. Model fit was assessed by visualising residuals and checking model convergence, singularity, multicollinearity and outliers. There was evidence of complete separation in SVD trials for whether the cat found the found the hidden toy and for which box the cat checked first. Random effects were removed to diagnose the variables causing separation issues and these variables were removed from the models. Multicollinear variables (VIF < 5) were also removed from the models and then backward elimination was carried out as outlined in *Regression models*. Odds ratios were calculated by exponentiating the log odds.

## Poisson models

Poisson models were fitted for frequencies of toy box-directed behaviours and empty box-directed behaviours. Full models had the same fixed and random effects as outlined in *Regression models*. Model fit was similarly assessed by visualising residuals and checking model convergence, singularity, multicollinearity and outliers, with additional checks for overdispersion and zero-inflation to ensure models were not under- or overfitting zeroes. There were no model fit concerns with toy box-directed behaviours. However, there were concerns with overdispersion for empty box-directed behaviours and so a negative binomial model was used. The fit of the negative binomial model was re-assessed and no longer had evidence of overdispersion. Incidence ratio ratios (IRRs) were calculated by exponentiating the coefficients. Final models of best fit can be found online at (https://doi.org/10.25377/sussex.28287896).

## Results

There were 10 female cats (*M* age = 4.0 years, *SD* age = 4.1 years, range = 0.5–11 years) and eight male cats (*M* age = 4.52 years, *SD* age = 3.33 years, range = 0.5–10 years) in our sampled population. All but two female cats were

neutered. Thirteen cats were mixed breed, and five cats were purebred (three Ragdoll, two Birman). Fourteen cats had outdoor access, and four cats were indoor-only cats.

The number of trials that cats displayed toy- and apparatus-directed behaviours are described in Table 2. Ten cats (56%) did not find the hidden toy in SVD trials, with eight out of these ten cats (80%) not attempting to find the toy at all. Conversely, there were 16 SVD trials where the cat found the toy, of which there were five trials where four cats did not touch the toy (three with the owner and two with the researcher). Mean latencies of toy manipulation and to first approach either the toy or empty box, in addition to durations of gazing at the bystander, in all experimental trials are presented in Table 3.

In SVD trials, cats who lived in single-cat household had longer relative durations of gazing at the box manipulator (M = .07, SE = .02) than cats who lived in dual-cat households (M = .03, SE = .01, $F_{1,30}$ = 4.44, p = .043, $\eta^2_G$ = .13, 95% CI 0.00–1.00). Additionally, cats had longer relative durations of gazing at the box manipulator if the toy was hidden in the left-hand side box of the manipulator (M = 0.07, SE = 0.03) in comparison to when the toy was hidden in the right-hand side box (M = 0.04, SE = 0.01, $F_{1,30}$ = 4.43, p = .043, $\eta^2_G$ = .11, 95% CI 0.00–1.00), suggesting a right-hand bias from the perspective of the cat. The starting distance of the cat also significantly influenced the relative duration of gazing at the box manipulator ($F_{3,30}$ = 4.55, p = .009, $\eta^2_G$ = .28, 95% CI 0.04–1.00), with pairwise comparisons demonstrating that cats who were 0 cm away from the apparatus had higher relative durations of gazing at the box manipulator (M = .13, SE = .06) in comparison to cats who were 16 cm away from the apparatus (M = .03, SE = .01, p = .005).

**Table 2. Number of trials in which cats found the toy, played with the toy, which box they checked first, their starting distance, the toy hiding box location, are reported in their respective SVD or event condition with the owner and researcher. The frequency of toy and empty box-directed behaviours are also reported per condition and frequencies of more than one have been grouped together for ease of interpretation (N = 18 cats).**

| Measures | Trial type | | | | | | |
|---|---|---|---|---|---|---|---|
| | Single visible displacement | | | Consistent event | | Violation event | |
| | Owner | Researcher | | Owner | Researcher | Owner | Researcher |
| **Toy hiding box location** | Left: 8 (44.44%) Right: 10 (55.56%) | Left: 6 (33.33%) Right: 12 (66.67%) | | Left: 8 (44.44%) Right: 10 (55.56%) | Left: 12 (66.67%) Right: 6 (33.33%) | Left: 6 (33.33%) Right: 12 (66.67%) | Left: 2 (11.11%) Right: 16 (88.89%) |
| **Box checked first** | Toy box: 8 (44.44%) Empty box: 5 (27.70%) No attempt: 5 (27.70%) | Toy box: 7 (38.89%) Empty box: 1 (5.56%) No attempt: 10 (55.56%) | | | | | |
| **Toy found** | Found: 10 (55.56%) Not found: 3 (16.67%) No attempt: 5 (27.78%) | Found: 6 (3.33%) Not found: 2 (11.11%) No attempt: 10 (55.56%) | Cat plays with toy | Yes: 10 (44.44%) No: 8 (55.56%) | Yes: 5 (27.78%) No: 13 (72.22%) | Yes: 8 (55.56%) No: 10 (44.44%) | Yes: 5 (27.78%) No: 13 (72.22%) |
| **Toy box-directed behaviour** | 0: 6 (33.33%) ≥ 1: 18 (66.67%) | 0: 7 (38.89%) ≥ 1: 11 (61.11%) | | 0: 6 (33.33%) ≥ 1: 12 (66.67%) | 0: 6 (33.33%) ≥ 1: 12 (66.67%) | 0: 10 (55.56%) ≥ 1: 8 (44.44%) | 0: 15 (83.33%) ≥ 1: 3 (16.67%) |
| **Empty box-directed behaviour** | 0: 10 (55.56%) ≥ 1: 8 (44.44%) | 0: 10 (55.56%) ≥ 1: 8 (44.44%) | | 0: 11 (61.11%) ≥ 1: 7 (38.89%) | 0: 17 (94.44%) ≥ 1: 1 (5.56%) | 0: 14 (77.78%) ≥ 1: 4 (22.22%) | 0: 16 (88.89%) ≥ 1: 2 (11.11%) |
| **Starting distance** | 0 cm: 2 (11.11%) 16 cm: 10 (55.56%) 32 cm: 4 (22.22%) ≥ 48 cm: 2 (11.11%) | 0 cm: 4 (22.22%) 16 cm: 8 (44.44%) 32 cm: 1 (5.56%) ≥ 48 cm: 5 (27.78%) | | 0 cm: 6 (33.33%) 16 cm: 7 (38.39%) 32 cm: 1 (5.56%) ≥ 48 cm: 4 (22.22%) | 0 cm: 3 (16.67%) 16 cm: 7 (38.89%) 32 cm: 1 (5.56%) ≥ 48 cm: 7 (38.89%) | 0 cm: 3 (16.67%) 16 cm: 6 (33.33%) 32 cm: 6 (33.33%) ≥ 48 cm: 3 (16.67%) | 0 cm: 2 (11.11%) 16 cm: 5 (27.78%) 32 cm: 4 (22.22%) ≥ 48 cm: 7 (38.89%) |

**Table 3. Mean durations of gazing at the box manipulator and mean latencies to play with the toy and first move towards the apparatus are reported in their respective SVD or event condition with the owner and researcher alongside standard errors and ranges (*N* = 18 cats).**

| Behaviour | Trial type | | | | | |
|---|---|---|---|---|---|---|
| | Single visible displacement | | Consistent event | | Violation event | |
| | Owner | Researcher | Owner | Researcher | Owner | Researcher |
| Mean duration of gazing at toy (s) | 9.37 (SE = 1.66) (*n* = 11) (1.68–16.67) | 8.43 (SE = 3.48) (*n* = 5) (0.68–16.56) | 7.88 (SE = 1.56) (*n* = 15) (0.80–20.64) | 8.70 (SE = 2.23) (*n* = 15) (0.44–26.96) | 4.83 (SE = 0.89) (*n* = 14) (0.64–11.12) | 6.70 (SE = 1.57) (*n* = 15) (0.28–19.84) |
| Mean duration of gazing at box manipulator (s) | 3.69 (SE = 2.94) (*n* = 9) (0.88–10.12) | 6.68 (SE = 1.81) (*n* = 10) (0.68–20.52) | 2.04 (SE = 0.59) (*n* = 9) (0.56–6.08) | 4.79 (SE = 2.33) (*n* = 6) (0.80–14.12) | 2.83 (SE = 0.73) (*n* = 9) (0.56–6.23) | 3.90 (SE = 1.09) (*n* = 5) (0.88–6.32) |
| Mean latency to play with toy (s) | 34.00 (SE = 6.81) (*n* = 7) (12.44–59.92) | 40.05 (SE = 9.88) (*n* = 4) (20.36–59.44) | 7.41 (SE = 2.44) (*n* = 10) (0.24–21.28) | 5.43 (SE = 1.79) (*n* = 8) (1.80–16.80) | 9.30 (SE = 5.29) (*n* = 5) (0.32–29.72) | 2.74 (SE = 0.99) (*n* = 5) (0.08–5.24) |
| Mean latency to first move towards apparatus (s) | 11.31 (SE = 4.26) (*n* = 13) (0–40.84) | 11.46 (SE = 2.74) (*n* = 12) (1.12–31.52) | 9.47 (SE = 2.27) (*n* = 11) (1.44–20.88) | 4.46 (SE = 1.24) (*n* = 9) (0.60–11.20) | 7.36 (SE = 2.88) (*n* = 8) (0.00–23.68) | 4.87 (SE = 3.28) (*n* = 4) (0.72–14.52) |

In SVD trials, no predictors in the final model significantly predicted which box the cats checked first (toy or empty box), including the first person to manipulate the boxes (OR = −0.13, 95% CI 0.00–1.36, *p* = .127), the manipulating person themselves (either owner or researcher) in the current trial (OR = 7.36, 95% CI 0.64–268.53, *p* = .165) and cat sex (OR = 10.80, 95% CI 0.85–439.27, *p* = .113). The full models did not outperform the null models in SVD trials for whether the cat found the toy and the relative duration of gazing at the toy, meaning that none of our predictors could predict these dependent variables.

In event trials, cats who were first presented with the consistent event had longer relative durations of gazing at the toy (M = .24, SE = .04) in comparison to cats who were first presented with the violation event (M = .14, SE = .03, $F_{1,68}$ = 6.16, *p* = .015, $\eta^2_G$ = .08, 95% CI 0.01–1.00). Cats had longer relative durations of gazing at the toy if the researcher presented the trials first (M = .26, SE = .04) in comparison to the owner presenting the trials first (M = .14, SE = .02, $F_{1,68}$ = 5.71, *p* = .019, $\eta^2_G$ = .07, 95% CI 0.00–1.00). Additionally, female cats had longer relative durations of gazing at the toy (M = .23, SE = .04) than male cats (M = .16, SE = .03, $F_{1,68}$ = 6.84, *p* = .010, $\eta^2_G$ = .09, 95% CI 0.01–1.00).

In event trials, as illustrated in Fig 2, whether the cat played with the toy was significantly affected by the trial type (consistent or violation), the first person to carry out the box manipulation (researcher or owner), household type (single- or dual-cat household), whether a cat had outdoor access (yes or no) and cat breed (purebred or mixed). Contrary to our expectations, the odds of playing with the toy were significantly higher in consistent trials than violation trials (OR = −0.20,

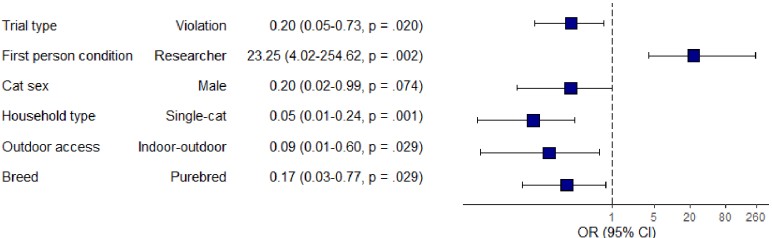

**Fig 2. Forest plot depicting experimental and lifestyle predictors for whether a cat played with the toy in event trials.** Significant predictors have *p* < .05. The squares and bars represent the odds ratios with 95% confidence intervals. Odds ratios calculated by exponentiating the log odds from multivariate logistic regression model.

95% CI 0.05–0.73, $p$ = .020) and significantly higher when the researcher acted as the box manipulator in comparison to the owner (OR = 23.25, 95% CI 4.02–254.62, $p$ = .002).

In event trials, cats had shorter latencies to play with the toy if they had outdoor access (M = 8.33, SE = 1.07 s) in comparison to indoor-only cats (M = 2.17, SE = .46 s, $F_{1,22}$ = 5.26, $p$ = .031, $\eta^2_G$ = .11, 95% CI 0.00–1.00). Cats also had shorter latencies to play with the toy if they were mixed breed (M = 4.81, SE = .70 s) in comparison to purebred (M = 15.56, SE = 2.69 s, $F_{1,22}$ = 19.44, $p$ < .001, $\eta^2_G$ = .42, 95% CI 0.16–1.00). The starting distance of the cat also significantly influenced the latency to play with the toy ($F_{3,22}$ = 4.16, $p$ = .017, $\eta^2_G$ = .21, 95% CI 0.00–1.00), with pairwise comparisons demonstrating that cats who were 0 cm away from the apparatus had shorter latencies to play with the toy (M = 2.16, SE = 0.65 s) than cats who were 16 cm away from the apparatus (M = 8.42, SE = 1.80 s, $p$ = .014). The full models did not outperform the null models in event trials for latency to move towards the apparatus and the relative duration of gazing at the manipulator.

In event trials, there were overall low frequencies of toy and empty box-directed behaviours. Cats were less likely to display toy box-directed behaviours in researcher trials than owner trials (IRR = 0.49, 95% CI 0.27–0.85, $p$ = .015). Additionally, cats were less likely to display toy box-directed behaviours in violation trials than consistent trials (IRR = 0.41, 95% CI 0.22–0.72, $p$ = .003). Outdoor access had no influence on the number of toy box-directed behaviours (IRR = 0.64, 95% CI 0.36–1.19, $p$ = 0.14). Similar to toy box-directed behaviours, cats were less likely to display empty box-directed behaviours in researcher trials than owner trials (IRR = 0.22, 95% CI 0.05–0.82, $p$ = .023). Additionally, cats were less likely to display empty box-directed behaviours when they started ≤ 48 cm from the apparatus in comparison to a close starting distance (IRR = 0.07, 95% CI 0.00–0.54, $p$ = .029).

## Discussion

In SVD trials, no experimental or lifestyle predictor significantly influenced which box the cats investigated first (toy or empty box), whether the cat found the toy during SVD trials and the duration of gazing at the toy itself. Cats would not have been able to gaze at the toy itself unless they had found the toy during SVD trials. Despite previous research concluding that cats readily solve both single (Stage 4) and successive (Stage 5) VD trials [18–20,22], our study found that cats found the toy in less than half of all SVD trials (44%) with many of these cats (42%) not finding the toy due to no attempts. Our findings are likely explained by the fact that subjects were difficult to motivate, a prevailing issue in domestic cat cognition research [66–68], as previous research repeatedly demonstrates that cats can reach Stage 5 object permanence [18,19,22]. Because person familiarity was not found to significantly influence any behaviours during SVD trials, it is unlikely that person familiarity can explain these relatively high numbers of no attempts. Perhaps the toy itself was not a salient enough reward to be incentivising for cats to move around the boxes to find the toy, despite offering a choice of toy prior to trials. Instead, a food reward with odour controls may have been more motivating for cats in our experimental set-up, of which previous research had found that cats persisted more with an object permanence task with a food reward when compared to the same task with a soft toy pillow as a reward [22].

Despite the low frequency of toy-directed behaviour in SVD trials, cats still differed in their gaze behaviours towards the box manipulator. Cats had longer gaze durations towards the manipulator if they lived in a single-cat household in comparison to a dual-cat household. The presence of the other resident cat may have acted as a distraction for the participating cat; when doors were closed so the other resident cat could not interfere with the current trial, or when the resident cat was in the same room (whichever the cat was most comfortable with), the participating cat often gazed towards the door or the other cat. Otherwise, cats also had longer gaze durations towards the manipulator if the toy was hidden in the left box of the manipulator (right side of the cat). Previous research using a progressive elimination task similarly found that cats were biased towards searching in an outer bowl on their right side [69]. However, further research will need to be carried out to investigate the extent of lateral biases in domestic cats. Furthermore, cats who were close to the apparatus had longer durations of gazing at the box manipulator (regardless of familiarity) than cats who were 16 cm away, suggesting that cats may use proximity as an indicator of general interest in events in their surroundings [70]. This is also supported

by our finding in event trials of which cats who were ≥ 48 cm away at the start of the observation interval were less likely to display empty box-directed behaviours than cats who were close to the apparatus.

Contrary to our prediction, we found that cats were less likely to display toy box-directed behaviour and play with the toy when presented with violation events in comparison to knowledge-consistent events. Cats were able to discriminate between knowledge-consistent and violation events but displayed more interest in the consistent events. Additionally, we found that cats gazed at the toy for a shorter amount of time when presented with the violation event before the consistent event. Previous research using violation-of-expectancy paradigms have demonstrated that human infants and dogs display longer gaze durations and more exploratory behaviours towards an object that seemingly defies expectations [35,46–48]. Empirically, the cats in our sample responded opposite to expectations formed from human children and dogs, and further research will be needed to discriminate between motivational and cognitive explanations for this puzzling response. As mentioned by a reviewer, it is unlikely, but not impossible, that cats were primed to expect a box switch from the SVD trials (of which cats were always shown the toy re-appearing out of the same box it was hidden) or from the randomised presentation order of the consistent and violation event trials.

We found that cats had longer gaze durations at the toy and were more likely to play with the toy when the researcher presented the event trials before the owner. However, in-line with our prediction, we found that cats were less likely to display toy and empty box-directed behaviours in researcher trials in comparison to owner trials. This presents somewhat conflicting findings; the cat is initially drawn to the novelty of the researcher's presence overall, yet the cat will maintain a perceived safe distance from the researcher in the current trial. Studies that investigate other aspects of cat cognition report that a number of cats are excluded due to apparent aversion to the researcher and their subsequent display of abnormal behaviour, for example, increased passivity [51]. Previous research has found that cats remain longer in proximity with an object that has been handled by their owner in comparison to a stranger [58], whereas other research has found that person familiarity does not influence behaviours such as duration of proximity or physical contact with humans [52]. In Strange Situation Tests [71], there is mixed evidence as to whether cats differentially change their proximity or play behaviours in the presence of strangers compared to the presence of their owner [55,56]. Otherwise, cats are able to pass A-not-B trials when ostensive cues are provided by a researcher (an unfamiliar person) but not the owner [57]. However, our findings show that cats display complex apparatus-directed behaviours under experimental circumstances, even in their home environment, in response to an unfamiliar person carrying out an experimental manipulation. Cats are still shown to have an interest in the experiment and may be encouraged to participate under appropriate circumstances, which may argue for the inclusion of cats who are shy or aversive to the presence of a researcher. However, these differences in toy-directed behaviour could also be explained by the cat's own individuality, which has been suggested to be the main contributing factor for how a cat may behave in the presence of an unfamiliar person, for example, with approaches, play, and distance from the person [53,59], all of which are highly relevant behaviours for engagement in our study.

We found that multiple lifestyle factors influenced playing with the toy during event trials. We found that female cats gazed for longer at the toy and were more likely to play with the toy than male cats. There is little empirical research on differences in gazing behaviour between male and female cats, although—contrary to our findings--males have been found to play more frequently than females [72]. Indoor-only cats were more likely to play with the toy but were slower to initiate play with the toy than indoor-outdoor cats. Although not ecologically relevant in terms of mimicking a hunting scenario, as in Dumas [29], playing with toys may have been more relevant for indoor-only cats, but perhaps less relevant for cats with outdoor access, who have additional enrichment from the outside environment. However, we did not record normal play behaviour between owners and their cats, so further research would be necessary to determine if baseline play exposure may increase or decrease interest in experimental paradigms involving toys as a reward. Mixed breed cats were more likely to play with the toy and initiated play with the toy faster than purebred cats. The purebred cats in our sample consisted of Birman and Ragdoll breeds with temperaments known to be gentle and sociable but not very playful, particularly in comparison to other purebreds such as Siamese or Russian Blue ([73], Cat Fanciers' Association,

CFA https://cfa.org/breeds/; The International Cat Association, TICA https://tica.org/). However, our sample included only five purebred cats in total (three Ragdoll, two Birman), so breed comparisons in toy-directed behaviours are limited by the small number of purebred cats in comparison to mixed breed cats. Finally, we found that cats from dual-cat households were more likely to play with the toy than cats from single-cat households. Competition over shared resources such as food or toys may occur in multi-cat households [74–76], although we did not assess inter-cat relationships as part of the present study, so this is a potential scope for future research.

After exclusions, we retained 18 cats in 6 trials each (108 trials in total). Seven cats were excluded from the initial pool of subjects, some for motivation or fear-related reasons. Our statistical analyses were limited by our small sample size (as indicated by the wide confidence intervals), and so further research should be carried out using larger samples to further our knowledge on how cats perceive consistent and violation events in the presence of familiar and unfamiliar people. Some variables which had no significant predictors, such as the first box the cat checked, had very wide confidence intervals, which indicate an increased chance of a Type II error [77], so could be reassessed in future studies with larger sample sizes. However, we acknowledge that recruiting and retaining house cats for cognitive studies is challenging. Furthermore, our exploratory analyses examined multiple lifestyle factors simultaneously and increased the likelihood of obtaining significant results by chance. Additionally, as we used two adjacent boxes, the short distance between the two potential hiding locations may not have warranted a highly different reaction based on the toy's re-appearance from either the toy or empty box, therefore inter-container distance should be manipulated in future (e.g., for chimpanzees: [78]; for dogs: [79]). As such, our study also falls under the criticism of associative learning [25], because cats may not have perceived the apparatus as having two separate hiding locations, so cats would have expected the toy to come out of the apparatus as a whole.

Another limitation that should be mentioned is that, by implementing a citizen science design, there was a lack of control over confounding variables such as the starting distance of the cat and the exact gaze and facial expression of the box manipulator. These design choices were made to encourage motivation and engagement throughout the trials. Although we attempted to control for some of these confounding variables by including them in the statistical models, readers should be made aware that such variables may have also interfered with our findings. Previous research has found that cats had more frequent approaches and longer durations of time in proximity with attentive humans versus passive humans [52,59], as well as longer durations of head rubbing and play behaviour with attentive versus passive humans [80], so the attentional display of the box manipulator may have encouraged cats to interact with the toy in some trials more than others, although this would require further investigation. Standardised protocols for interacting with cats in cognitive research are yet to be established, so, due to the cats' general disposition for avoiding strangers and being easily disturbed by changes in their environment, we chose to maintain as much of the normalcy of the home environment as possible during the trials. However, some confounding variables, such as the left-right position of the hiding box, are capable of being controlled in future studies.

This study was designed for administration in cats' home environments. We had expected that our cat-friendly adaptations would foster recruitment, but this did not turn out to be the case—uptake of our social media recruitment efforts was relatively low. We also expected that cats would be more motivated to participate by allowing flexible starting distances and freedom of movement during trials. However, a substantial minority (42%) of these cats made no apparent effort to search for the toy in the simplest presentation (SVD trials). Even more surprisingly, cats had longer gaze durations at toys that re-appeared from the container in which it had been hidden than to toys that appeared after invisible displacement in the IVD trials; this is opposite from the most typical pattern reported in children and dogs. Thus, in our study, cats discriminated congruent from incongruent appearances in their gaze behaviour, but in the direction opposite expectations from other species. This warrants further investigation for whether this can be attributed to motivational or cognitive explanations; that is, from the standpoint of object permanence, it is not clear from the present study whether our sample is unrepresentative of cats or whether cats are unrepresentative of mammals in this paradigm. Finally, notwithstanding the

relatively small sample, we found several lifestyle factors influenced cats' responses, including sex, breed, and presence of other cats in the household. This suggests that cats might be substantially more sensitive to contextual factors than other domesticated species.

## Acknowledgments

We would like to thank all of the owners and their cats who took the time to participate in this study.

## Author contributions

**Conceptualization:** Jemma Forman, David A. Leavens.

**Data curation:** Jemma Forman.

**Formal analysis:** Jemma Forman.

**Investigation:** Jemma Forman.

**Methodology:** Jemma Forman, David A. Leavens.

**Project administration:** Jemma Forman, David A. Leavens.

**Software:** Jemma Forman, Jordan S. Rowe.

**Supervision:** David A. Leavens.

**Validation:** Jordan S. Rowe.

**Visualization:** Jemma Forman.

**Writing – original draft:** Jemma Forman.

**Writing – review & editing:** Jemma Forman, Jordan S. Rowe, David A. Leavens.

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
