## [Decision Letter · Decision Letter 0]

PONE-D-24-43874Object permanence in domestic cats (Felis catus) using violation-of-expectancy by owner and strangerPLOS ONE

Dear Dr. Forman,

Thank you for submitting your manuscript to PLOS ONE. After careful consideration, we feel that it has merit but does not fully meet PLOS ONE’s publication criteria as it currently stands. Therefore, we invite you to submit a revised version of the manuscript that addresses the points raised during the review process.

We look forward to receiving your revised manuscript.

Kind regards,

Mary Diane Clark, PhD

Academic Editor

PLOS ONE

Journal requirements:    When submitting your revision, we need you to address these additional requirements. 1. Please ensure that your manuscript meets PLOS ONE's style requirements, including those for file naming. The PLOS ONE style templates can be found at https://journals.plos.org/plosone/s/file?id=wjVg/PLOSOne_formatting_sample_main_body.pdf and https://journals.plos.org/plosone/s/file?id=ba62/PLOSOne_formatting_sample_title_authors_affiliations.pdf 2. Please match your authorship list in your manuscript file and in the system. 3. Please include a caption for figure 1a and 2.  4. Your ethics statement should only appear in the Methods section of your manuscript. If your ethics statement is written in any section besides the Methods, please move it to the Methods section and delete it from any other section. Please ensure that your ethics statement is included in your manuscript, as the ethics statement entered into the online submission form will not be published alongside your manuscript. 

Additional Editor Comments:

As you can see, you have a variety of reviews. I think that they have given you some important feedback. Please respond to their concerns.Also please review the formatting for the paper. References are not APA for this journal.

Reviewers' comments:

Reviewer's Responses to Questions

**Comments to the Author**

1. Is the manuscript technically sound, and do the data support the conclusions?

Reviewer #1: No

Reviewer #2: Yes

Reviewer #3: Yes

2. Has the statistical analysis been performed appropriately and rigorously? 

Reviewer #1: Yes

Reviewer #2: Yes

Reviewer #3: Yes

3. Have the authors made all data underlying the findings in their manuscript fully available?

Reviewer #1: Yes

Reviewer #2: Yes

Reviewer #3: Yes

4. Is the manuscript presented in an intelligible fashion and written in standard English?

Reviewer #1: Yes

Reviewer #2: Yes

Reviewer #3: Yes

5. Review Comments to the Author

Reviewer #1: This study provides valuable data on the procedures often used in home-based experiments with cats, examining object permanence abilities and the impact of whether the experimenter is the owner or a stranger. However, as noted in the comments below, I found it difficult to assess the study because the authors' logic on several key points was not entirely clear.

Major points

1�I understand that the research question is whether cats can pass Stage 6 of object permanence. If this is the case, it seems that the Event trials implement the Invisible Displacement procedure of Stage 6. From the perspective of the cats, it could be perceived that the boxes were being swapped in both Consistent and Violation events. Therefore, if cats possess the cognitive ability to pass Stage 6, then Violation events may not actually be violations but rather consistent. I found it difficult to evaluate the paper as I did not fully understand the authors’ logic on this point. Since the results differed from the authors' predictions, reinterpreting the results based on the logic I have suggested would amount to HARKing, which should be avoided. However, I would like the authors to clarify this point thoroughly.

2. In addition to the main factors of whether the manipulator was the owner or the researcher, and whether the event was a Violation or Consistent, the authors have also comprehensively examined factors such as the cat’s age, sex, multi-cat household, outdoor access, and breed. However, when examining numerous factors simultaneously, there is a risk of obtaining significant results by chance. Since no predictions were made for factors beyond the primary factors, these should be considered exploratory analyses, making it difficult to draw definitive conclusions from the results of this experiment.

3. I noticed that "citizen science" is listed as a keyword, which implies procedural issues, as owners were involved in conducting the experiment. Unlike conventional experiments where factors are tightly controlled, I have concerns about the lack of such control here. Specifically, there was no control over the distance between the box and the cat, the behavior of the cat during Event trials, or pseudorandomization of the target box’s left-right position. These uncontrolled factors may introduce too many potential confounds affecting the experiment.

Minor points

1. Lines 20-204 state that low-density foam was adhered inside the boxes to prevent auditory cues. However, cats have more acute hearing than humans. Was it experimentally confirmed that no auditory cues were actually produced?

2. Around Line 219, two boxes are presented side by side to the cats. What was the distance between the boxes? (Apologies if I missed this detail.)

3. Line 230 notes that the bystander was present in the same room during the experiment. What was their approximate distance from the cat? Since this likely varied by room setup, even a range would be helpful.

4. Lines 237 & 248 state that the manipulator "sat neutrally" after the manipulation. What about their gaze and facial orientation?

5. Lines 243-244: As previously mentioned, was the cat allowed to move freely while the manipulator conducted the operation?

6. Line 258: It does not appear to be explicitly stated that the 72 Event trials are divided between the manipulator being either the owner or the researcher.

7. It was unclear whether left and right are described from the perspective of the cat or otherwise.

8. Line 507: The authors mention the small sample size as a limitation, but was any effect size or post-hoc power analysis conducted?

Reviewer #2: This is an interesting study on cat behavior. The manuscript was in good condition. The discussion was adequatley based on the results, and the authors drew conclusions in line with the data,

However, the findings were not linked to an overall discuss of how these results improved our understanding of object permanence. Essentially, the final paragraph of the discussion should be on the same rough level as the first paragraph of the introduction.

Reviewer #3: Overview and general recommendation:

This manuscript covers an interesting topic: the object permanence, that is the understanding that objects continue to exist while out of sight. The lack of knowledge and studies in this field on cats requires further research. Thereby, the paper will fit into a gap in knowledge about feline cognition. The article is clearly laid out and all the elements are present, referring to most relevant literature in the area. The authors made a very good job identifying strengths and new insights to be explored with further studies and limitations that need to be considered. For me the paper, a part for a very few subsequent elements, could be accepted in the present form.

Specific comments

Abstract

-Please check the length of the abstract: the Plos One guideline gives a maximum limit of 300 words.

-For those who are not in the field, it is not clear the paradigm of the permanence of the object compared to the sensorimotor period of a human infant (Piaget, 1954). Perhaps it would be useful to give a very brief explanation of this without taking it for granted.

Introduction

-Please add the title “Introduction” heading at the beginning of the introduction section

-Lines 100-125: I would probably move this part in the discussion: it is fine just to mention the reason for your choice, but the argument of discussion seems too broad in this session, and I would prefer to move it into the discussion session

Materials and Methods

-Lines 150-175: are these cats included in the experiment?

-Lines 187-191: I would move this part in the result section, because you “didn’t decide” these data but they are the random result of the sampling, right?

Discussion

-Lines 494-499: in your result you find that mixed breed cats were more likely to play with the toy and initiated play with the toy faster than purebred cats. You rightly supposed that temperaments of the purebred cats represented in your sample (Birman and Ragdoll, that are known to be gentle and sociable but not very playful) could be the reason for your result. But it is important underling that you have just 5 purebred cats and this makes the comparison not so effective.

References

Please check the format of the reference list; below is reported a part of the guideline for authors:

“Formatting references

Because all references will be linked electronically as much as possible to the papers they cite, proper formatting of references is crucial.

PLOS uses the reference style outlined by the International Committee of Medical Journal Editors (ICMJE), also referred to as the “Vancouver” style. Example formats are listed below. Additional examples are in the ICMJE sample references.

A reference management tool, EndNote, offers a current style file that can assist you with the formatting of your references. If you have problems with any reference management program, please contact the source company's technical support.

Journal name abbreviations should be those found in the National Center for Biotechnology Information (NCBI) databases.

Source Format

Published articles

Hou WR, Hou YL, Wu GF, Song Y, Su XL, Sun B, et al. cDNA, genomic sequence cloning and overexpression of ribosomal protein gene L9 (rpL9) of the giant panda (Ailuropoda melanoleuca). Genet Mol Res. 2011;10: 1576-1588.

Devaraju P, Gulati R, Antony PT, Mithun CB, Negi VS. Susceptibility to SLE in South Indian Tamils may be influenced by genetic selection pressure on TLR2 and TLR9 genes. Mol Immunol. 2014 Nov 22. pii: S0161-5890(14)00313-7. doi: 10.1016/j.molimm.2014.11.005.”

-Line 59: The reference “(Holekamp, 2007)” is missing in the reference list.

-Line 65: The reference “Dumas (1992)” is missing in the reference list.

- Line 108: The reference “(Vitale & Udell, 2019)” is missing in the reference list.

- Lines 160-161: The reference “(Pyari et al., 2023)” is missing in the reference list.

-the subsequent references are missing in the text:

1. Chijiiwa, H., Takagi, S., Arahori, M., Anderson, J. R., Fujita, K., & Kuroshima, H. (2021a). Cats (Felis catus) show no avoidance of people who behave negatively to their owner. Animal Behavior and Cognition, 8(1), 23-35. https://doi.org/10.26451/abc.08.01.03.2021

2. Chijiiwa, H., Takagi, S., Arahori, M., Hori, Y., Saito, A., Kuroshima, H., & Fujita, K. (2021b). Dogs and cats prioritize human action: choosing a now-empty instead of a still-baited container. Animal Cognition, 24, 65-73. https://doi.org/10.1007/s10071-020-01416-w

3. Koo, T. K., & Li, M. Y. (2016). A guideline of selecting and reporting intraclass correlation coefficients for reliability research. Journal of Chiropractic Medicine, 15(2), 155– 163. https://doi.org/10.1016/j.jcm.2016.02.012

4. Majecka, K., & Pietraszewski, D. (2018). Where’s the cookie? The ability of monkeys to track object transpositions. Animal Cognition, 21, 603-611. https://doi.org/10.1007/s10071-018-1195-x

5. McHugh, M. L. (2012). Interrater reliability: the kappa statistic. Biochemia Medica, 22(3), 276-282. https://doi.org/10.11613/BM.2012.031

6. Schober, P., Boer, C., & Schwarte, L. A. (2018). Correlation coefficients: appropriate use and interpretation. Anesthesia & Analgesia, 126(5), 1763–1768. https://doi.org/10.1213/ANE.0000000000002864

Figures

Please add the title in the figures

6. PLOS authors have the option to publish the peer review history of their article (what does this mean? ). If published, this will include your full peer review and any attached files.

**Do you want your identity to be public for this peer review?** For information about this choice, including consent withdrawal, please see our Privacy Policy .

Reviewer #1: No

Reviewer #2: No

Reviewer #3: No

---

## [Author Response · Author response to Decision Letter 1]

28 Feb 2025

We thank the Editor and all reviewers for providing thoughtful improvements for this manuscript. We have revised the manuscript to fit the journal requirements of PLOS ONE. We appreciate the concerns regarding the expectations of the box switching and have further clarified some aspects of the study design that were identified by reviewers. We have carefully considered and responded to each of the comments in turn.

Journal Requirements

Point 1: Please ensure that your manuscript meets PLOS ONE's style requirements, including those for file naming. The PLOS ONE style templates can be found at

Response: Apologies for these formatting oversights. We have checked the templates from the above links and have now formatted the manuscript to meet PLOS ONE’s style requirements for file naming.

Point 2: Please match your authorship list in your manuscript file and in the system.

Response: The authorship list is now formatted to meet PLOS ONE’s style requirements and is matched in the system.

Point 3: Please include a caption for figure 1a and 2.

Response: The figure captions were included in the original manuscript immediately after the first paragraph in which the figure is cited but have now been formatted for PLOS ONE standards. Because this comment has been similarly noted by Reviewer 3, we are hoping this is satisfactory.

I had previously referred to the submission guidance: “Figure captions must be inserted in the text of the manuscript, immediately following the paragraph in which the figure is first cited (read order). Do not include captions as part of the figure files themselves or submit them in a separate document.”

Point 4: Your ethics statement should only appear in the Methods section of your manuscript. If your ethics statement is written in any section besides the Methods, please move it to the Methods section and delete it from any other section. Please ensure that your ethics statement is included in your manuscript, as the ethics statement entered into the online submission form will not be published alongside your manuscript.

Response: The ethics statement has now been moved to the Methods section of the manuscript.

At the request of the PLOS ONE office, we have additionally updated the DOI that links to our data and supplementary materials to reflect the PLOS ONE guidance for online repositories (we previously used Mendeley Data, but we have now moved our data to an approved repository of figshare). Please note the DOI will become active once approved by my institution.

Reviewer Comments

Major points

Point 5: I understand that the research question is whether cats can pass Stage 6 of object permanence. If this is the case, it seems that the Event trials implement the Invisible Displacement procedure of Stage 6. From the perspective of the cats, it could be perceived that the boxes were being swapped in both Consistent and Violation events. Therefore, if cats possess the cognitive ability to pass Stage 6, then Violation events may not actually be violations but rather consistent. I found it difficult to evaluate the paper as I did not fully understand the authors’ logic on this point. Since the results differed from the authors' predictions, reinterpreting the results based on the logic I have suggested would amount to HARKing, which should be avoided. However, I would like the authors to clarify this point thoroughly.

Response: We predicted that cats would show more interest in the toy (longer gaze durations, more physical touches with the toy) when it re-appeared out of the box it was not seen to be originally hidden (violation event) in comparison to when it re-appeared out of the box it was seen to be originally hidden (consistent event). We maintain this prediction in the manuscript based on the findings of previous literature on human infants and domestic dogs and to avoid HARKing. However, contrary to our prediction, we found that cats showed more interest in the toy in consistent events in comparison to violation events.

Because of these contradictory findings, the reviewer provided an alternative explanation that the cats might have assumed the boxes were switched in both conditions (consistent and violation events), hence why the cats showed more interest in the consistent events when the boxes were not switched. We lack a ready explanation for why the cats would have made such an assumption, but this is a valid alternative interpretation.

Before the event trials, cats were shown a SVD trial performed by either the owner or the researcher. This SVD trial would not have primed the cats to assume a box switch. If a cat did not independently search for the toy in the allotted time, the box manipulator would always reveal the toy coming out of the same box that it was hidden in for SVD trials. Furthermore, the presentation order of either the consistent or violation event was randomised, so it is unlikely that cats would have had expectations about the location of the toy based on previous presentations. Moreover, the cats had longer gaze durations at the toy when presented with the consistent event before the violation event (which does lend evidence to the hypothesis that cats were expecting a box switch). However, because this analysis was grouped across both event types, the relative proportion of gazing at the toy in either the consistent or violation conditions is unclear. For example, a cat may have had longer gaze durations at the toy when presented with the consistent event first, but that cat may have gazed for longer durations during the violation trials, even if they were presented after the consistent trials.

As we see it, the fundamental facts in play, here, are (a) that the cats did discriminate between consistent and violation conditions in their gaze durations and toy touching, (b) that the pattern is opposite to that reported for dogs and young children, who also discriminate the conditions, albeit in the opposite direction (tend to engage with the object in the violation conditions), and (c) nobody directly measures object permanence as a cognitive event; rather, there is a longstanding tendency to interpret discriminations in terms of object representations.

It is worth noting that if the cats assumed switching, this would constitute evidence for Stage 6 object permanence, at the cost of an objective assay for the capacity—that is, gazing longer at the toy in both the consistent (cats) and violation (humans, dogs) conditions would be taken, variously, as evidence for object permanence, thus invalidating the entire protocol.

We did not have any definitive evidence for cats to either have a) expected a switch, or b) expected a non-switch. It is theoretically possible that they could have expected a switch, but this would have far-reaching implications for the whole protocol. We did not want to burden the manuscript with this whole explanation but remain keen to revise further if the editor or reviewer require it. We have no direct way to address this concern raised by the reviewer in the current manuscript.

Nonetheless, in the manuscript itself, we have included a brief sentence that addresses this valid point raised by the reviewer.

Lines 449 - 453 “As mentioned by a reviewer, it is unlikely, but not impossible, that cats were primed to expect a box switch from the SVD trials (in which cats were always shown the toy re-appearing out of the same box it was hidden) or from the randomised presentation order of the consistent and violation event trials.”

Point 6: In addition to the main factors of whether the manipulator was the owner or the researcher, and whether the event was a Violation or Consistent, the authors have also comprehensively examined factors such as the cat’s age, sex, multi-cat household, outdoor access, and breed. However, when examining numerous factors simultaneously, there is a risk of obtaining significant results by chance. Since no predictions were made for factors beyond the primary factors, these should be considered exploratory analyses, making it difficult to draw definitive conclusions from the results of this experiment.

Response: We have now clarified that our analyses of lifestyle factors were carried out as exploratory analyses throughout the manuscript. As this reviewer correctly explains, we have now also included the limitation that there is a risk of obtaining significant results by chance by examining numerous factors simultaneously.

Lines 120 – 122 “Furthermore, we also carried out exploratory analyses for whether demographic factors (such as cat age, cat sex, living in a multi-cat household, outdoor access, or breed) influenced any observed behaviours during SVD and event trials.”

Lines 284 – 287 “For both SVD and event trials, we carried out exploratory analyses for numerous lifestyle factors (cat age, cat sex, multi-cat household, outdoor access, and breed) and their effects on our outcome variables.”

Lines 508 – 510 “Furthermore, our exploratory analyses examined multiple lifestyle factors simultaneously and increased the likelihood of obtaining significant results by chance.”

Point 7: I noticed that "citizen science" is listed as a keyword, which implies procedural issues, as owners were involved in conducting the experiment. Unlike conventional experiments where factors are tightly controlled, I have concerns about the lack of such control here. Specifically, there was no control over the distance between the box and the cat, the behavior of the cat during Event trials, or pseudorandomization of the target box’s left-right position. These uncontrolled factors may introduce too many potential confounds affecting the experiment.

Response: This reviewer is correct that citizen science may introduce procedural issues (such as the distance between the boxes as mentioned in Point 9 (originally: Minor Point 2). Owners were verbally instructed on how to move the boxes in real-time by the researcher during the trials, but this still allowed room for minor procedural errors. We have added a sentence to clarify that owners were verbally instructed what to do during the trials as this information was previously missing.

Lines 204 – 206 “Before the trials, the researcher physically demonstrated to the owner how they should carry out the study in relation to moving the boxes, the toy and the folder. The researcher also verbally instructed the owner what to do during the trials.”.

It is also true that citizen science experiments do not have as tight controls over extraneous variables, which in our study would include variables such as the starting distance of the cat. This was implemented with the hope this would have encouraged the cat to engage in experimental trials. A lack of motivation has been a prevailing challenge for cat cognition research (Chijiwa et al., 2021a, 2021b; Pagé & Dumas, 2009), of which citizen science provides a feasible alternative research method which encourages motivation and participation in experimental trials.

We have now explained this limitation of citizen science methods in the Discussion section. We hope this provides a fairer representation of both the benefits and limitations of citizen science research.

Lines 517 – 523 “Another limitation that should be mentioned is that, by implementing a citizen science design, there was a lack of control over confounding variables such as the starting distance of the cat. This design choice was made to encourage motivation and engagement throughout the trials. Although we attempted to control for some of these variables in statistical analysis (limited by the small sample size), readers should be made aware that confounding variables may have also interfered with our findings. Some confounding variables, such as the left-right position of the hiding box, are capable of being controlled in future studies.”

Lines 524 – 530 “This study was designed for administration in cats’ home environments. We had expected that our cat-friendly adaptations would foster recruitment, but this did not turn out to be the case—uptake of our social media recruitment efforts was relatively low. We also expected that cats would be more motivated to participate by allowing flexible starting distances and freedom of movement during trials. However, a substantial minority (42%) of these cats made no apparent effort to search for the toy in the simplest presentation (SVD trials).”

Minor points

Point 8: Lines 20-204 state that low-density foam was adhered inside the boxes to prevent auditory cues. However, cats have more acute hearing than humans. Was it experimentally confirmed that no auditory cues were actually produced?

Response: The low-density foam was adhered to the boxes to reduce the sound from the toy being placed inside the box. It was not experimentally confirmed that no auditory cues were produced. As this reviewer correctly points out, because cats have more acute hearing than humans, it is not accurate to conclude that sounds cues were not produced at all. The word ‘prevent’ was not an accurate description of the purpose of the low-density foam and instead we have replaced this with ‘reduce’.

Lines 180 – 181 “The researchers glued low density foam (1/2” thickness) to the inside of the boxes to help to reduce sound cues from the toy being placed inside.”

Point 9: Around Line 219, two boxes are presented side by side to the cats. What was the distance between the boxes? (Apologies if I missed this detail.)

Response: In order for the two boxes to be completely hidden behind an A4-sized folder, the two boxes were positioned next to one another. Owners were instructed to position the boxes a couple of centimetres apart, but some owners positioned the boxes together, so they were physically touching. Provided that both boxes were hidden behind the folder, the boxes could either have no physical distance between them, or a very small distance of only a couple of centimetres separating them, but this was not standardised in the current study. This detail was not previously reported in the Methods section but has been made clearer in the resubmitted manuscript.

Lines 178 – 180 “The two boxes were instructed to be positioned next to each other and to be separated by a small distance (approx. 2 cm).”

In the discussion section, we acknowledge that this is a limitation of our study design because cats may not have perceived the two boxes as two separate hiding locations.

Point 10: Line 230 notes that the bystander was present in the same room during the experiment. What was their approximate distance from the cat? Since this likely varied by room setup, even a range would be helpful.

Response: We agree with this reviewer that an approximate range for where the bystander was sitting would be helpful for readers to understand the overall setup of the experiment. We have included two sentences in the Methods section to clarify the distance of the bystander.

Lines 211 – 213 “Depending on the layout and size of the room, the bystander would sit between 1 m to 3 m away from the apparatus. The bystander and the box manipulator would always swap positions, so the bystander was always in the same place for every trial.”

Point 11: Lines 237 & 248 state that the manipulator "sat neutrally" after the manipulation. What about their gaze and facial orientation?

Response: Thank you for calling attention to this wording. We have now refined this description of the manipulator to better describe their attentive state during the trials. Because the manipulator was allowed to look toward and talk to the cat to encourage motivation, we have removed the word ‘neutral’ to avoid misinterpretation. We have more accurately labelled the manipulator as attentive instead of neutral when describing the procedure in the Single visible displacement trials and Knowledge-consistent event trials subheadings.

Lines 220 – 222 “The manipulating person then sat for 60 s and waited for the trial to end. During this time, the manipulating person was attentive to the cat (i.e. facing

---

## [Decision Letter · Decision Letter 1]

PONE-D-24-43874R1Object permanence in domestic cats (Felis catus) using violation-of-expectancy by owner and strangerPLOS ONE

Dear Dr. Forman,

Thank you for submitting your manuscript to PLOS ONE. After careful consideration, we feel that it has merit but does not fully meet PLOS ONE’s publication criteria as it currently stands. Therefore, we invite you to submit a revised version of the manuscript that addresses the points raised during the review process. 

Thank you for the work addressing reviewers comments. In addition, the explanation of your changes was extremely helpful. The first reviewer has two remaining concerns that once completed will allow the paper to be accepted. The second comment asks for a power analysis and an effect size to be included in your results.  These two pieces of information will allow readers to better understand not only the statistical significance but the real world significance of your findings.

We look forward to receiving your revised manuscript.

Kind regards,

Mary Diane Clark, PhD

Academic Editor

PLOS ONE

Journal Requirements:

Additional Editor Comments:

Thank you for all of the changes in this revision. The first reviewer would like to see two additional changes. When you complete those, the paper can be accepted.

Reviewers' comments:

Reviewer's Responses to Questions

**Comments to the Author**

1. If the authors have adequately addressed your comments raised in a previous round of review and you feel that this manuscript is now acceptable for publication, you may indicate that here to bypass the “Comments to the Author” section, enter your conflict of interest statement in the “Confidential to Editor” section, and submit your "Accept" recommendation.

Reviewer #1: (No Response)

Reviewer #3: All comments have been addressed

2. Is the manuscript technically sound, and do the data support the conclusions?

Reviewer #1: Yes

Reviewer #3: Yes

3. Has the statistical analysis been performed appropriately and rigorously? 

Reviewer #1: No

Reviewer #3: Yes

4. Have the authors made all data underlying the findings in their manuscript fully available?

Reviewer #1: Yes

Reviewer #3: Yes

5. Is the manuscript presented in an intelligible fashion and written in standard English?

Reviewer #1: Yes

Reviewer #3: Yes

6. Review Comments to the Author

Reviewer #1: The authors have responded sincerely to most of my questions and comments.

However, I would like to request two further improvements regarding their revisions and responses.

For my question, “Lines 237 & 248 state that the manipulator ‘sat neutrally’ after the manipulation. What about their gaze and facial orientation?”, the authors addressed it by adding information about the manipulator’s behavior during the trials. However, if their description is accurate, the manipulator’s behavior could have had an effect on the results. This should be acknowledged in the discussion as a limitation.

For another question, “Line 507: The authors mention the small sample size as a limitation, but was any effect size or post-hoc power analysis conducted?”, they responded that they did not conduct such analyses. However, effect size and post-hoc power analysis can be performed even with a small sample size. The results of these analyses should be provided for the readers.

Reviewer #3: The Authors made a good job addressing my comments. For me the paper could be accepted in the present form.

7. PLOS authors have the option to publish the peer review history of their article (what does this mean? ). If published, this will include your full peer review and any attached files.

**Do you want your identity to be public for this peer review?** For information about this choice, including consent withdrawal, please see our Privacy Policy .

Reviewer #1: No

Reviewer #3: No

---

## [Author Response · Author response to Decision Letter 2]

30 May 2025

We thank the Editor and reviewers for providing further improvements for this manuscript. We agree that gaze and facial orientation may have exerted an effect on the results and we have now included this in the Discussion section. We also agree that effect sizes should still be reported alongside our analyses to provide more information on the suitability of the sample size. We have responded to the remaining comments below.

Due to saving errors, our track changes were inadvertently removed from the track changes document. We have instead highlighted the track changes in yellow. Apologies for this inconvenience.

Reviewer Comments

Reviewer #1: The authors have responded sincerely to most of my questions and comments.

However, I would like to request two further improvements regarding their revisions and responses.

1. For my question, “Lines 237 & 248 state that the manipulator ‘sat neutrally’ after the manipulation. What about their gaze and facial orientation?”, the authors addressed it by adding information about the manipulator’s behavior during the trials. However, if their description is accurate, the manipulator’s behavior could have had an effect on the results. This should be acknowledged in the discussion as a limitation.

Response: Thank you for this further suggestion. We also acknowledge that gaze and facial orientation may have had an effect on the results and have now included this in the discussion as a limitation. Our reasoning for this was to promote the normalcy of the home environment and to encourage participation and interaction with the apparatus. An unfamiliar person in the cat’s home environment, with the additive influence of that stranger behaving in an unexpected way (e.g., sitting still with no discernible facial expression), would likely have further discouraged the cat from participating in the study. We also note that standardised protocols for interacting with cats during cognitive studies are yet to be established, although there is available research that shows that cats are more interactive with attentive humans (in comparison to passive humans). We have now included this information in the Discussion section.

Lines 527 – 543 “Another limitation that should be mentioned is that, by implementing a citizen science design, there was a lack of control over confounding variables such as the starting distance of the cat and the exact gaze and facial expression of the box manipulator. These design choices were made to encourage motivation and engagement throughout the trials. Although we attempted to control for some of these confounding variables by including them in the statistical models, readers should be made aware that such variables may have also interfered with our findings. Previous research has found that cats had more frequent approaches and longer durations of time in proximity with attentive humans versus passive humans [59, 52], as well as longer durations of head rubbing and play behaviour with attentive versus passive humans [80], so the attentional display of the box manipulator may have encouraged cats to interact with the toy in some trials more than others, although this would require further investigation. Standardised protocols for interacting with cats in cognitive research are yet to be established, so, due to the cats’ general disposition for avoiding strangers and being easily disturbed by changes in their environment, we chose to maintain as much of the normalcy of the home environment as possible during the trials. However, some confounding variables, such as the left-right position of the hiding box, are capable of being controlled in future studies.” 

2. For another question, “Line 507: The authors mention the small sample size as a limitation, but was any effect size or post-hoc power analysis conducted?”, they responded that they did not conduct such analyses. However, effect size and post-hoc power analysis can be performed even with a small sample size. The results of these analyses should be provided for the readers.

Response:

Effect sizes

Thank you for this suggestion. We agree that effect size calculations can be performed on linear regression models and have now reported this in the revised manuscript. Please note that odds ratios and incidence rate ratios were already calculated for logistic and Poisson regression models, respectively.

For effect sizes, generalised eta squared values were calculated for linear regression models. This is recommended for repeated measures designs (Bakeman, 2005; Cohen, 1988, 2013).

The newly-reported effect sizes are reported in the Response to Reviewers document (which are also reported in the revised manuscript text next to their respective analysis). There is a clear trend for wide confidence intervals (spanning from 0.00 – 1.00). This means that there is an increased chance of a Type II error occurring (Lee, 2016) and that we did not have a sufficient number of cats in our sample to have precise estimates. In larger sample sizes, the confidence intervals are typically narrower, and the interpretation is generally more reliable, even if the p value remains constant (Lee, 2016). Although our study found significant differences and small to medium effects between some of the variables (as shown by the p values of < .05 and effect sizes of < .26), the interpretation of the significance level is not reliable (as shown by the wide confidence intervals). This provides statistical support for the limitation of a small sample size, which we had already suspected, and means that some variables with no significant predictors (such as which box the cat checked first) may have a different result if reassessed in future studies.

Lines 311– 313 “Effect sizes were calculated using generalized eta squared (η2G). Effect sizes were interpreted as small (> .02 x < .13), medium (> .13 x < .26), or large (x > .26) [64, 65].”

Lines 511 – 517 “Our statistical analyses were limited by our small sample size (as indicated by the wide confidence intervals), and so further research should be carried out using larger samples to further our knowledge on how cats perceive consistent and violation events in the presence of familiar and unfamiliar people. Some variables which had no significant predictors, such as the first box the cat checked, had very wide confidence intervals, which indicate an increased chance of a Type II error [77], so could be reassessed in future studies with larger sample sizes. However, we acknowledge that recruiting and retaining house cats for cognitive studies is challenging.”

Post hoc analysis

We have reviewed the literature for theoretical perspectives on post hoc power analysis and have also consulted with statisticians at our institution. We do not wish to refute a reviewer comment, however, we also do not want to ignore the vast amount of literature that argues that post hoc power analyses are an unsuitable way to assess statistical power, and is not generally recommended for research (Heckman et al., 2022; Hoenig & Heisey, 2011; Levine & Ensom, 2001). There are some key differences in how population and sample means and standard deviations are calculated between a priori and post hoc power analysis, and because we already know our population parameters, this can have significant consequences on the resultant power statistic (Zhang et al., 2019). A statistical test will always have low observed power when non-significant effects are reported (Goodman & Berlin, 1994), so calculating the post hoc power does not provide any more information when we already know which variables are (non-)significant (Heckman et al., 2022). Furthermore, these differences can be even more exaggerated when using smaller sample sizes (Zhang et al., 2019), which was observed in the current study.

Even though the literature strongly suggests that post hoc power analyses should not be used to understand statistical power and sample size suitability, we would like to acknowledge that a priori power analyses are an ideal way to assess power and are important for interpreting results in the context of effect sizes, power, and p values. To strengthen any projects that we may carry out in the future, we will endeavour to have an a priori sample size calculation carried out before any data collection takes place.

We are hoping this response is satisfactory for addressing the underlying concern that our models were underpowered due to small sample sizes, and that the reporting of the effect sizes and confidence intervals have provided more context to the interpretation of our results.

References:

Bakeman, R. (2005). Recommended effect size statistics for repeated measures designs. Behavior Research Methods, 37, 379-384.

Cohen, J. (1988). The effect size. Statistical power analysis for the behavioral sciences. Abingdon: Routledge, 77-83.

Cohen, J. (2013). Statistical power analysis for the behavioral sciences. Routledge.

Goodman, S. N., & Berlin, J. A. (1994). The use of predicted confidence intervals when planning experiments and the misuse of power when interpreting results. Annals of internal medicine, 121(3), 200-206.

Heckman, M. G., Davis, J. M., & Crowson, C. S. (2022). Post hoc power calculations: an inappropriate method for interpreting the findings of a research study. The Journal of Rheumatology, 49(8), 867-870.

Hoenig, J. M., & Heisey, D. M. (2001). The abuse of power: the pervasive fallacy of power calculations for data analysis. The American Statistician, 55(1), 19-24.

Lee, D. K. (2016). Alternatives to P value: confidence interval and effect size. Korean journal of anesthesiology, 69(6), 555.

Levine, M., & Ensom, M. H. (2001). Post hoc power analysis: an idea whose time has passed?. Pharmacotherapy: The Journal of Human Pharmacology and Drug Therapy, 21(4), 405-409.

Zhang, Y., Hedo, R., Rivera, A., Rull, R., Richardson, S., & Tu, X. M. (2019). Post hoc power analysis: is it an informative and meaningful analysis?. General psychiatry, 32(4), e100069.

---

## [Editor Report · Decision Letter 2]

Object permanence in domestic cats (Felis catus) using violation-of-expectancy by owner and stranger

PONE-D-24-43874R2

Dear Dr. Forman,

We’re pleased to inform you that your manuscript has been judged scientifically suitable for publication and will be formally accepted for publication once it meets all outstanding technical requirements.

Kind regards,

Mary Diane Clark, PhD

Academic Editor

PLOS ONE

Additional Editor Comments (optional):

Thank you for these additional edits. I appreciate the care with which you responded to the latest comments.
---

## [Editor Report · Acceptance letter]

PONE-D-24-43874R2

PLOS ONE

Dear Dr. Forman,

I'm pleased to inform you that your manuscript has been deemed suitable for publication in PLOS ONE. Congratulations! Your manuscript is now being handed over to our production team.

Kind regards,

on behalf of

Dr. Mary Diane Clark

Academic Editor

PLOS ONE